# Bilateral Cervical Lymphadenopathy after mRNA COVID-19 Vaccination on Oral Squamous Cell Carcinoma Patient: A Case Report

**DOI:** 10.3390/diagnostics12071518

**Published:** 2022-06-21

**Authors:** Eun-Sung Kang, Moon-Young Kim

**Affiliations:** Department of Oral and Maxillofacial Surgery, College of Dentistry, Dankook University, 119 Dandae-ro, Cheonan 31116, Chungcheongnam-do, Korea; petiteye@naver.com

**Keywords:** mRNA COVID-19 vaccine, cervical lymphadenopathy, oral squamous cell carcinoma

## Abstract

We report the case of a 59-year-old man with squamous cell carcinoma (SCC) of the right mandibular gingiva, who presented with bilateral cervical lymphadenopathy (CLA) after mRNA coronavirus disease 2019 (COVID-19) vaccination. The patient was diagnosed. Imaging studies performed prior to surgery revealed bilateral CLA and axillary lymphadenopathy (LA) ipsilateral to the vaccination site. Fine-needle aspiration (FNA) biopsy of the left CLA revealed reactive lymph nodes. The patient underwent surgical intervention for the malignant tumor, and the specimen was sent for histopathologic evaluation. The biopsy-proven cancer stage was pT3N0Mx. Positron emission tomography (PET-CT), performed six months after surgery, showed persistent bilateral CLA. However, FNA of the left axillary LA once again showed no evidence of metastasis or recurrence. Since the treatment plan may change based on the type of LA, it is important to figure out whether an mRNA vaccine has been administered to patients with head and neck cancer.

## 1. Introduction

There has been continuous development of vaccines against the coronavirus disease 2019 (COVID-19), and mRNA vaccines are being widely used in practice. However, critical side effects of mRNA vaccines, such as reactive lymphadenopathy, have been reported [1,2,3,4,5]. Although axillary LA is common after vaccine administration, those of the supraclavicular and cervical regions have also been found in some patients [1,2,3,4,5]. mRNA vaccine-induced reactive LA may be misdiagnosed as lymphatic metastasis on imaging tests [6]. Since the number of lymphatic metastases is an important factor in determining the stage and prognosis of head and neck cancer, careful examination by oral and maxillofacial surgeons who treat head and neck cancer is required.

To date, LA has been reported only on the ipsilateral side of the vaccine administration site. We herein document a case in which LA was also found on the contralateral side of the vaccine administration site, mimicking bilateral cervical lymph-node metastases of head and neck cancer.

## 2. Case Report

A 59-year-old man, who had previously received first and second doses of the Moderna mRNA COVID-19 vaccine (Moderna Therapeutics, Cambridge, MA, USA) on his left shoulder on 30 June and 28 July, respectively, visited the department of Oral and Maxillofacial Surgery (OMS) with complaint of an ulcerative lesion on the mandibular right gingiva. As the lesion showed an exophytic, erythroleukoplakic appearance, incisional biopsy was performed, and the lesion was diagnosed as SCC (Figure 1). The patient was admitted to the department of OMS on 7 August for surgical treatment of the mandibular gingival SCC. Imaging studies performed prior to the scheduled surgery, such as enhanced computer tomography (CT) (Figure 2), magnetic resonance (MR) imaging (Figure 3a,b), and PET-CT (Figure 4a,b), revealed hypermetabolic enlarged lymph nodes in the bilateral cervical levels Ib and II. Hypermetabolic enlarged lymph nodes were also found in left axillary levels I to III (Figure 5). The primary malignant lesion was limited to the mandibular right gingiva without any signs of local or distant metastasis, and preoperative FNA of the left cervical lymph node revealed reactive LA (Figure 6). Only small lymphoid cells were detected at the suspicious left lymph node.

Following segmental mandibulectomy, right-sided supraomohyoid neck dissection (SOHND-cervical levels Ia to III), and fibular free flap (FFF) under general anesthesia, the primary carcinoma lesion and neck dissection specimen were sent for histopathologic examination. Histopathologically, the margin of the primary lesion was clear, and there were no metastatic lymph nodes on right side of the neck (pT3N0M0). The patient was discharged without any complications.

The PET-CT performed five months after surgery revealed a persistent reactive LA at the left cervical level II (Figure 7), while axillary LA had been resolved (Figure 8). Histopathological examination of a fresh FNA biopsy specimen from several cervical lymph nodes revealed small lymphoid cells without any evidence of metastatic carcinoma (Figure 9). The post-operative PET-CT images revealed no evidence of residual pathology or infection, and the patient was also completely free of relevant clinical symptoms upon follow up.

## 3. Discussion

In patients with head and neck cancer, lymphatic metastasis is an important prognostic factor for determining postoperative chemoradiotherapy [7]. As mRNA vaccine-related CLA can mimic the behavior of metastatic lymph nodes in patients with head and neck cancer, it is important to take special care.

LA is an acute or chronic inflammatory process of the lymph nodes that occurs in response to a variety of pathogenic agents. Cases of reactive LA on the ipsilateral side of vaccine injection have been reported prior to the development of mRNA COVID-19 vaccines [8]. COVID-19 vaccines are being actively administered, and reactive LA on the injected side is one of the side effects [9]. For example, axillary reactive lymph nodes are easily found after vaccination [1,2]. According to Wolfson S. et al., the patients demonstrated LA as early as 1 day after the first dose of mRNA vaccine and as late as 71 days after the second dose [10]. Additionally, there are reports of ipsilateral CLA after mRNA COVID-19 vaccination [4]. mRNA vaccine-related LA may mimic metastatic lymph nodes, making the diagnosis difficult for oncologists. In relation to mRNA COVID-19 vaccine, the highest number of reactive lymph nodes was found after taking the Moderna vaccine [9]. Furthermore, the incidence of LA after the second dose was significantly higher compared to that after the first dose [9]. Because our patient had received second doses of the Moderna vaccine, the possibility of reactive LA was high. LA is known to be a common side effect of mRNA COVID-19 vaccination that usually disappears within a short period of time. However, there have also been long-term LA cases reported to be persisting up to 10 weeks [11] or over 43 weeks following vaccination [10].

Similar to metastatic lymph nodes, reactive LA appears as a hypermetabolic lesion on PET-CT. Therefore, it can be easily misdiagnosed as metastatic lymph nodes [12]. The existence, number, and location (ipsilateral or bilateral) of metastatic cervical lymph nodes are important to determine the prognosis in head and neck cancer. Head and neck SCC usually causes ipsilateral lymph node metastasis. However, in rare cases, contralateral lymph node metastasis can occur due to crossing afferent lymphatic vessels, tumor spread along the midline, or extensive involvement of the ipsilateral lymph nodes [13]. In addition, it can occur when there is no real midline barrier, as in certain anatomical areas [14].

In this case, hypermetabolic enlarged lymph nodes were identified in bilateral cervical levels Ib and II on PET-CT. Moreover, hypermetabolic enlarged lymph nodes were identified in left axillary levels I to III. The tumor was restricted to the right mandibular gingiva, and massive lymph-node metastasis was not found ipsilaterally. Therefore, we hypothesized that the left CLA was reactive CLA caused by the mRNA COVID-19 vaccine, and lymph-node biopsy was required to prevent unnecessary surgery. Five months after surgery, PET-CT showed the remained reactive lymph node on same anatomic region, and there was no evidence of any other infection or pathologic condition on head and neck area. As there have been reports of long-lasting LA after mRNA COVID-19 vaccination, we concluded that LA which the patients had been shown was CLA due to mRNA COVID-19 vaccine [10,11].

To diagnose whether the suspicious lymph node is reactive or metastatic, the proper biopsy method should be chosen. One of the methods, such as open biopsy, sentinel lymph node biopsy, and FNA can be considered. Open biopsy involves complete or partial removal of the lymph node. Sentinel lymph nodes are the first lymph nodes that a malignant tumor may spread to and are determined by injecting a radioactive liquid into the region of the malignant tumor. The liquid moves through the lymphatic vessels and into the lymph nodes. The first nodes that the tracer drains into are the sentinel nodes. Surgical removal of the sentinel lymph nodes is performed and the specimen is sent for the histopathologic study. Sonogram-guided FNA is performed by inserting a fine needle into the lymph node to remove fluid and cells and the specimen is sent for the histopathologic study. FNA biopsy of reactive retro jugular lymph nodes have been reported [15]. As the anatomic location of the reactive lymph nodes of our patient was close to the retropharyngeal space, sentinel lymph node biopsy was thought to be dangerous. Therefore, FNA biopsy was performed.

To the best of our knowledge, this case report documents the first case of bilateral CLA following mRNA COVID-19 vaccination. However, some limitations regarding the preoperative differential diagnosis of the SCC lesion exist. Recent technological advancements have substantially improved the diagnostic accuracy of traditional adjunctive methods, such as the CT, MR, PET-CT, and FNA of the suspicious LA, as performed for our patient in this report. However, the diagnostic yield of our case remains limited owing to the absence of ultrasonographic (US) studies as part of the diagnostic process. Ultra-high frequency ultrasound (UHUFS), a novel ultrasonographic tool, has shown superior sensitivity, specificity, positive and negative predictive values for the detection of oral SCC lesions compared to those of conventional US in a previous study [16]. Another prior literature has also documented a strong correlation between UHUFS and post-operative histological findings, such as the tumor thickness and depth of invasion of oral SCC [17]. However, we feel that this obvious limitation is understandable as several other modalities previously confirmed to have high diagnostic yields were performed and, thus, the sufficient evidence of malignancy existed prior to surgery.

## 4. Conclusions

In this case, contralateral LA was observed in a patient with oral carcinoma. MR and PET-CT imaging findings mimicked those of metastatic lymph nodes in cancer patients. Since the treatment plan, including administration of adjuvant chemotherapy and/or radiotherapy after surgery, may change based on the type of LA, it is important to check whether an mRNA vaccine has been administered to patients with head and neck cancer. As mRNA vaccine may be prevailed the post COVID-19 era, it has become critical for oncologists to take a detailed vaccination history to prevent misdiagnose the cancer stage.

## Figures and Tables

**Figure 1 diagnostics-12-01518-f001:**
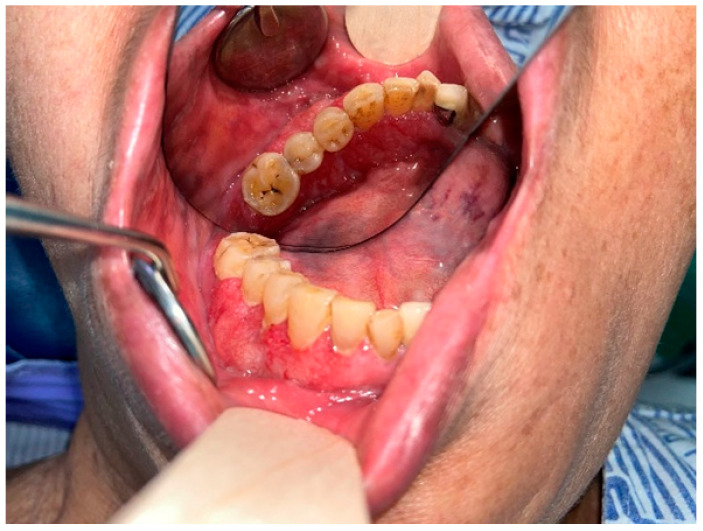
Clinical Photo of Suspected Lower Gum Lesion.

**Figure 2 diagnostics-12-01518-f002:**
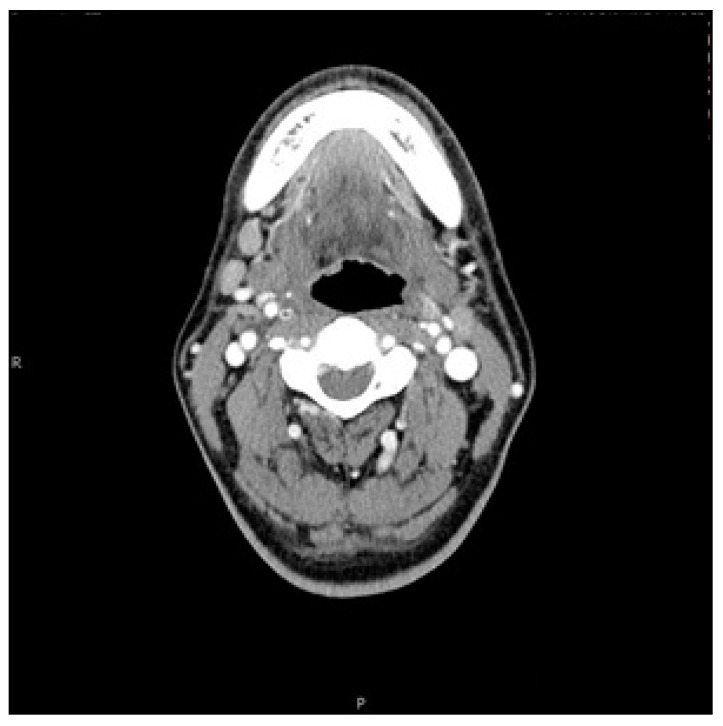
Axial view of right CLA on enhanced CT imaging.

**Figure 3 diagnostics-12-01518-f003:**
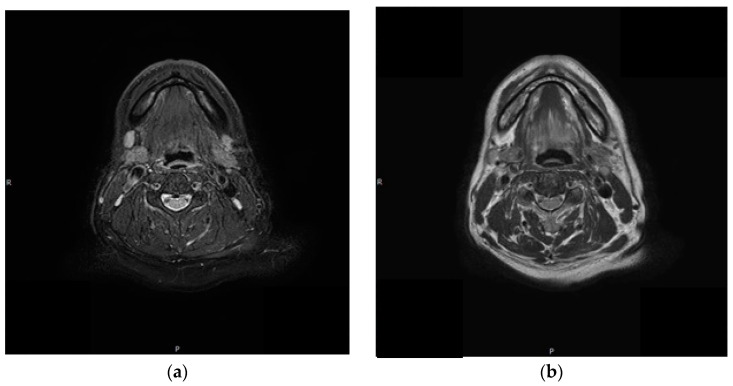
Axial view of CLA on MR imaging. (**a**) Right CLA; (**b**) Left CLA.

**Figure 4 diagnostics-12-01518-f004:**
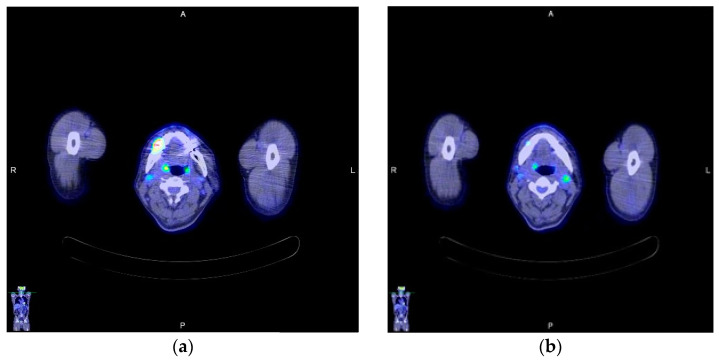
Axial view of CLA on PET–CT imaging: (**a**) Right CLA; (**b**) Left CLA.

**Figure 5 diagnostics-12-01518-f005:**
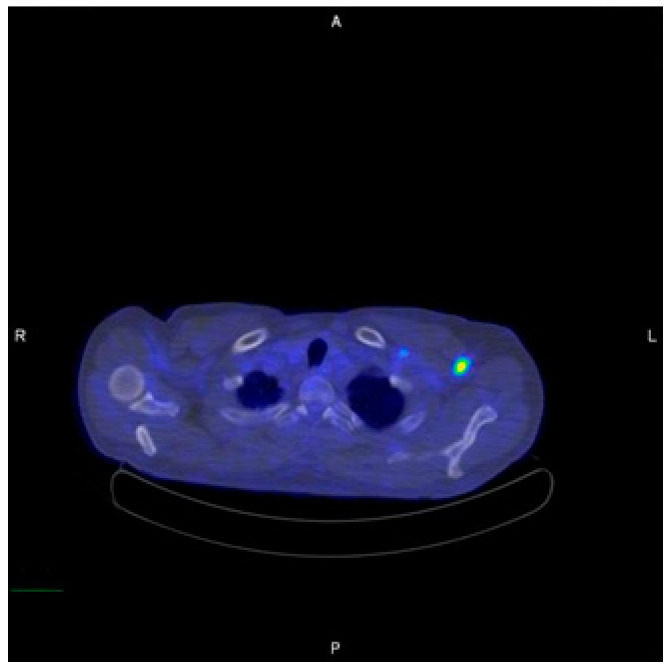
Pre–operation Axillary LA on PET–CT imaging.

**Figure 6 diagnostics-12-01518-f006:**
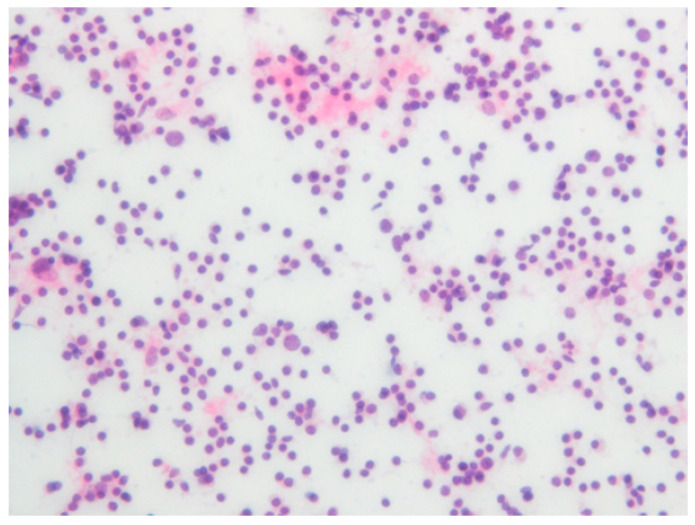
Pre–operation histopathologic imaging of FNA on left CLA (H&E stain).

**Figure 7 diagnostics-12-01518-f007:**
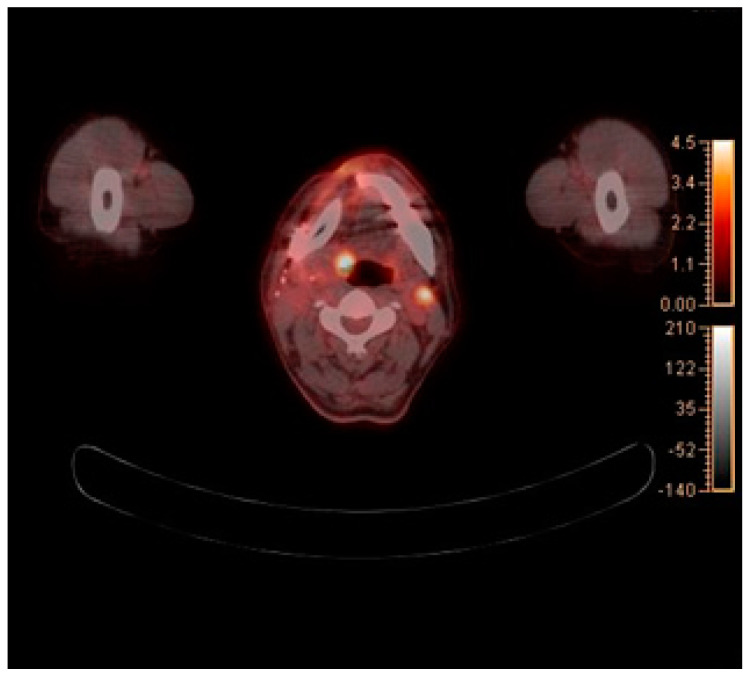
Axial view of CLA on PET–CT imaging 5 months after surgery.

**Figure 8 diagnostics-12-01518-f008:**
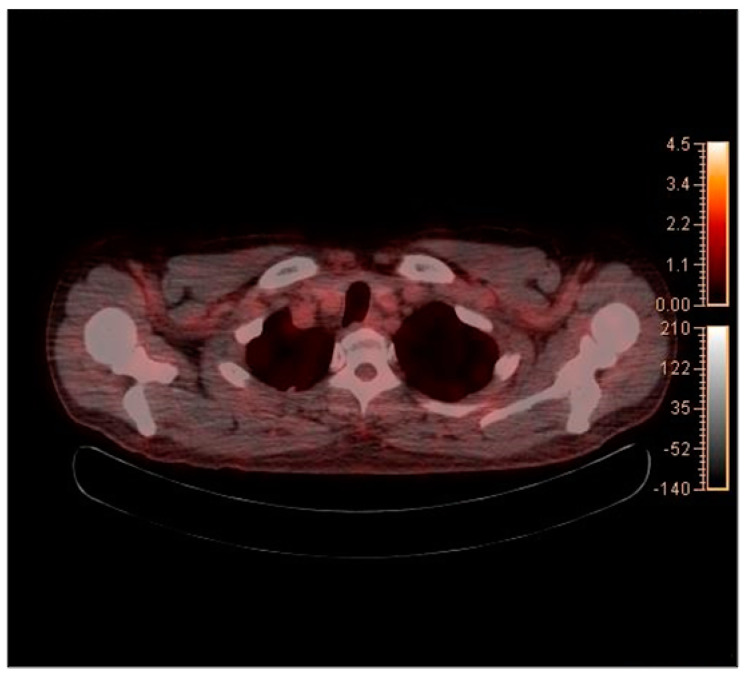
Axial view of axillar on PET–CT imaging 5 months after surgery.

**Figure 9 diagnostics-12-01518-f009:**
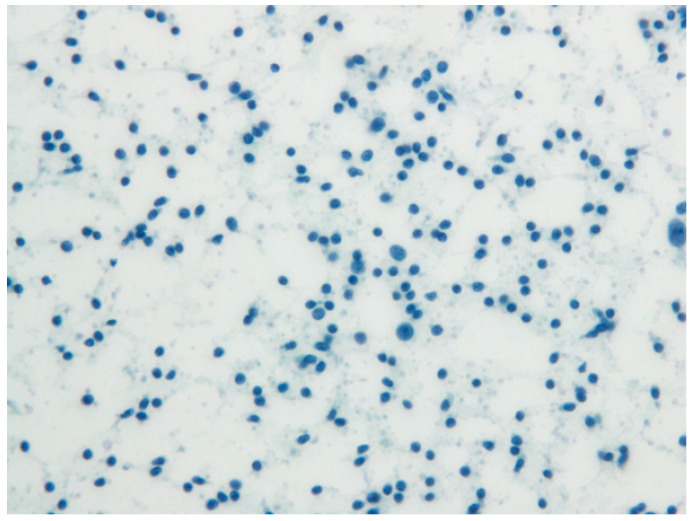
Post–operation histopathologic imaging of FNA on left CLA (Papanicolaou stain).

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
