# Peer review of "Bilateral Cervical Lymphadenopathy after mRNA COVID-19 Vaccination on Oral Squamous Cell Carcinoma Patient: A Case Report"

_diagnostics, 2022, doi:10.3390/diagnostics12071518_

Round 1
Reviewer 1 Report
Dear authors,
the article is well written satisfying all appropriate guidelines and can give important information for the development of the topic of mRNA-COVID-19 vaccinations of immunity system.
Regards
Author Response
Reviewer 1. Major comments
Reviewer #1: the article is well written satisfying all appropriate guidelines and can give important information for the development of the topic of mRNA-COVID-19 vaccinations of immunity system.
Response: We thank you for reviewing our manuscript.
Reviewer 2 Report
This is an interesting case however I feel there are some issues that hamper to establish a conclusive diagnosis regarding the cervical lymphadenopathy being related to the vaccine.
Indeed, Moderna vaccine has been associated with an increased risk of lymphadenopathy, mainly axillary, but this typically occurs within 2 to 4 days after vaccination and that regresses over time. The cervical and axillary lymphadenopathy reported herein were diagnosed many weeks after the second dose. The axillary lymphadenopathydisappeared during follow-up; therefore, it could be hypothesized that this could be due to the vaccine that endured longer than usual. However, the reactive cervical lymphadenopathy, which persisted, might be related to other infections in the head and neck region. It is not clear if the authors have performed a meticulous examination of the neck region and investigated any previous symptoms or signs suggestive of infection affecting the throat, ear, salivary glands, skin, or surrounding structures. I believe the discussion should address such limitations.
Author Response
Reviewer 2. Major comments
Reviewer #2: This is an interesting case however I feel there are some issues that hamper to establish a conclusive diagnosis regarding the cervical lymphadenopathy being related to the vaccine. Indeed, Moderna vaccine has been associated with an increased risk of lymphadenopathy, mainly axillary, but this typically occurs within 2 to 4 days after vaccination and that regresses over time. The cervical and axillary lymphadenopathy reported herein were diagnosed many weeks after the second dose. The axillary lymphadenopathydisappeared during follow-up; therefore, it could be hypothesized that this could be due to the vaccine that endured longer than usual. However, the reactive cervical lymphadenopathy, which persisted, might be related to other infections in the head and neck region. It is not clear if the authors have performed a meticulous examination of the neck region and investigated any previous symptoms or signs suggestive of infection affecting the throat, ear, salivary glands, skin, or surrounding structures. I believe the discussion should address such limitations.
Response: We appreciate the reviewer for valuable advice and detailed suggestions. The changes to the manuscripts are highlighted in the red letter. [Page 4, Line 74-76] We haved added that the patient hadn’t suffered any clinical infectious symptom around head and neck region, and on PET-CT imaging, there was no evidence of any infection or pathologic condition which may occur reactive lymphadenopathy. [Page 6, Line 93-95] According to Wolfson S et al., they reported that the patients demonstrated lymphadenopathy as early as 1 day after first dose and as late as 71 days after second dose. Also, incidence of reactive lymphadenopathy after second dose was higher compared to that after the first dose.[9] [Page 6, Line 101-104] According to Wolfson S et al., few patients suffered reactive lymphadenopathy persisted longer than 43 weeks[10], and Garreffa E et al. reported it can be persisted up to 10 weeks.[11] [Page 6, Line 118-123] Therefore our patient had been demonstrated lymphadenopathy on same anatomic region before the surgery and only five months past, it is suitable to say that the lymphadenopathy on PET-CT imaging is related to mRNA COVID-19 vaccination.
Reviewer 3 Report
The manuscript reports a case of lymphadenopathy following COVID-19 vaccination in a patient with a diagnosis of oral squamous cell carcinoma.
the diagnostic work-up is unclear. After identifying the lesion, was an incisional biopsy performed?
Why ultrasonography of the lesion was not performed?
Please provide clinical images of the lesion
The correlation with COVID-19 vaccination is debatable. Presumably, there is no link between the two conditions described. I would suggest the authors to limit the reporting to the case of squamous cell carcinoma.
In the study limitations, please report the absence of ultrasonographic assessment.
Author Response
Reviewer 3. Major comments
Reviewer #3: The manuscript reports a case of lymphadenopathy following COVID-19 vaccination in a patient with a diagnosis of oral squamous cell carcinoma. the diagnostic work-up is unclear. After identifying the lesion, was an incisional biopsy performed? Why ultrasonography of the lesion was not performed? Please provide clinical images of the lesion. The correlation with COVID-19 vaccination is debatable. Presumably, there is no link between the two conditions described. I would suggest the authors to limit the reporting to the case of squamous cell carcinoma. In the study limitations, please report the absence of ultrasonographic assessment.
Response: Thank you for reviewing our manuscript and for providing us with your valuable comment. [Page 1, Line 43] We described that after identifying the intraoral lesion, incisional biopsy had been performed. On tongue cancer ultrasonography can be helpful method to evalutate the thickness of the oral cancer. However our patient suffered lower gum cancer, furthermore the patient had been undergone imaging study such as enhanced CT, MR imaging, PET-CT to evaluate the size of the tumor. We thought further ultrasonographic assessment was not necessary. [Page 2, Figure 1] We have included clinical photo of the patient’s primary lower gum lesion. Primary cancer was limited to right side of the oral cavity and the patient showed bilateral reactive cervical lymphadenopathy on pre-operation imaging study. In rare cases, the oral cavity cancer can cause contralateral lymph node metastasis, but it can occur when there is no real midline barrier.[14] To identify whether the left side lymphadenopathy is reactive or metastatic before the surgery, the FNA was performed and the result was the reactive lymph node. During operation, the neck dissection of the right side was performed and there were non of metaststic lymph nodes. Furthermore, after 5 month reactive lymph nodes were still remained and patients had not suffered any infectious symptoms. Also on PET-CT imaging, there were no evidence of infection. Therefore, as there are studies that mRNA COVID-19 vaccine related lymphadenopathy can persist over 10 weeks [11], and 43 weeks [10], it is suitable to say that lymph nodes which were shown in imaging were reactive lymphadenopathy derived from Moderna vaccination.
Round 2
Reviewer 2 Report
The authors have satisfactory replied to my questions.
Author Response
Reviewer 2. Major comments
Reviewer #2: The authors have satisfactory replied to my questions.
Response: We are appreciated and thanks for your positive comment
Reviewer 3 Report
The manuscript has improved. However, the authors should list as a limitation the lack of use of ultrasonography, which is increasingly relevant for the evaluation of both oral SCC lesions and lymph nodes. The authors should check the following studies.
- doi: 10.1016/j.oooo.2019.09.012
- doi: 10.3390/app11167647
Author Response
Reviewer 3. Major comments
Reviewer #3: The manuscript has improved. However, the authors should list as a limitation the lack of use of ultrasonography, which is increasingly relevant for the evaluation of both oral SCC lesions and lymph nodes. The authors should check the following studies.
- doi: 10.1016/j.oooo.2019.09.012
- doi: 10.3390/app11167647
Response: Thank you for your comment and review. I reviewed the journals you had recommended to check. The limitation of our case report is that we didn’t perform the ultrahigh frequency ultrasound (UHUFS). As it is novel diagnostic tool which shows strong correlation with histological findings, and high specificity, sensitivity, positive predictive value and negative predictive value, what if UHUFS had been performed at preoperative process, much more precisely differential diagnosis of the SCC lesion would be done. Once again thank you for your kind recommendation.